# Minimum recommended micronutrient intake status and associated factors among pastoralist children aged 6–23 months in Aysaita district, Afar Region, Ethiopia, 2024: A community based cross-sectional study

**Ali Mohammed Ibrahim[1], Abdulkerim Hassen Moloro[2]\*, Dereje Desalegn Boshe[1], Yeshimebet Ali Dawed[3]**

**1** Department of Public Health, College of Medicine and Health Sciences, Samara University, Samara, Ethiopia, **2** Department of Nursing, College of Medicine and Health Sciences, Samara University, Samara, Ethiopia, **3** Department of Nutrition and Dietetics, School of Public Health, College of Medicine and Health Sciences, Wollo University, Dessie, Ethiopia

\* habdulkerim4@gmail.com

## Abstract

### Background

Micronutrient deficiencies are among the most prevalent public health problems in developing countries like Ethiopia. Despite government efforts to address these deficiencies, progress remains slow, leaving many children to endure serious health consequences. There is a scarcity of studies examining the minimum recommended micronutrient intake among children aged 6–23 months in this study area.

### Objective

This study aimed to assess minimum recommended micronutrient intake status and associated factors among pastoralist children aged 6–23 months in Aysaita District, Afar Region, Ethiopia, 2024.

### Methods

A community-based cross-sectional study design was conducted among 614 children aged 6–23 months with mothers from August 1–30/2024. Multi-stage sampling technique was used to select study participants. Data were collected by KoboToolbox using an interview-based structured questionnaire. The collected data were exported to Excel and then into STATA version 17 software packages. Descriptive statistics such as proportion, mean, median, cross-tabulation, and frequencies were calculated and presented in tables. Bivariable and multivariable logistic regression analyses at

**Data availability statement:** All relevant data are within the paper and its Supporting information files.

**Funding:** The author(s) received no specific funding for this work.

**Competing interests:** The authors have declared that no competing interests exist.

**Abbreviations:** ANC, Antenatal Care; AOR, Adjusted Odds Ratio; CI, Confidence Interval; COR, Crude Odds Ratio; CSA, Central Statistical Agency; EDHS, Ethiopian Demographic and Health Survey; FAO, Food and Agriculture Organizations; FMoH, Federal Ministry of Health; MDD, Minimum Dietary Diversity; MN, Micronutrients; MNP, Multiple Micronutrient Powder; NGO, Non-Governmental Organization; PNC, Postnatal Care; SDG, Sustainable Development Goals; USD, United States Dollar; VA, Vitamin A; VAS, Vitamin A supplements; WHO, World Health Organization.

a 95% confidence interval (CI) and a P-value of less than 0.05 in the multivariable logistic regression were considered significant predictors.

## Results

All study participants were included for data collection by giving a response rate of 100%. In this study, only 35.67% of the children aged 6–23 months had received at least one minimum recommended micronutrient intake status (35.67%; 95% CI: 31.96, 39.54%). Being a male child (AOR = 2.04; 95% CI: 1.42, 2.92), having media exposure (AOR = 2.50; 95% CI: 1.34, 4.64), having exclusive breast feeding (AOR: 5.09, 95% CI: 2.72, 9.52), being in the income category 7.07–35.34 USD (AOR: 2.90, 95% CI: 1.49, 5.65) and spontaneous vaginal delivery (AOR: 2.71, 95% CI: 1.20, 6.10) were factors associated with the minimum recommended micronutrient intake status.

## Conclusions and recommendations

This study revealed low adequate micronutrient intake among children aged 6–23 months, with only slightly over one-third (35.67%) meeting the minimum recommended status. Being a male child, having media exposure, having exclusive breast feeding, being in the income category 7.07–35.34 USD and spontaneous vaginal delivery were factors associated with the minimum recommended micronutrient intake status. To effectively address micronutrient intake, interventions must be more targeted and intensified. This involves strengthening programs that educate mothers on optimal infant and young child feeding, from exclusive breastfeeding to preparing diverse, nutrient-rich foods. Public messaging via community campaigns and media should be expanded to promote affordable, local food options and must explicitly target caregivers of girls to eliminate preferential feeding practices. Finally, to overcome economic barriers, initiatives like women's economic empowerment, social safety nets, and homestead food production (e.g., kitchen gardens, poultry) should be supported to ensure consistent household access to nutritious food.

## Introduction

The initial two years of life are a critical period for development, significantly influencing immune and physiological functions [1]. Despite this, children are highly vulnerable to undernutrition during this time. Key drivers include inadequate dietary intake, limited food access, nutritional taboos, and infectious diseases [2]. Contributing factors are multifaceted, encompassing socioeconomic and household characteristics such as low income [3], limited healthcare access [4], food insecurity [5], large family size [6], inadequate dietary diversity of the child [7] and poor sanitation [8] as well as child-specific issues like suboptimal breastfeeding practices [9], infections (e.g., diarrhea, malaria) [10], low birth weight [7], and inadequate complementary feeding [11].

Maternal factors, including low education [8], poor nutritional knowledge [12], and low body weight [7], also play a critical role.

In Ethiopia, micronutrient deficiencies among children are a major public health issue [13]. The most prevalent deficiencies include iron, iodine, vitamin A, and zinc [14]. These deficiencies are responsible for over half of all deaths in children under five worldwide [15]. Additionally, they contribute to weakened immunity, intellectual disabilities, reduced nutrient absorption, and delayed or impaired physical, mental, and psychomotor development [16–18].

Between 1990 and 2017, Ethiopia experienced a decrease in dietary deficiencies, with iron deficiency dropping by 20.1%, vitamin A deficiency by 16.7%, and iodine deficiency by 91.6% [19]. The National Nutrition Program (NNP II) is dedicated to implementing nutrition interventions aimed at ending hunger by 2030, including efforts to address and prevent micronutrient deficiencies [20]. Despite these governmental efforts, key indicators of micronutrient intake among young children remain low. For example, only 29% of children aged 6–23 months consumed vitamin A-rich foods, and 24% consumed iron-rich foods [21]. Additionally, just 6% of children aged 6–59 months received iron supplementation, while 53% received vitamin A supplementation [21]. Consequently, micronutrient deficiencies continue to be a significant issue in the country [19].

According to the 2016 Ethiopia Demographic and Health Survey (EDHS), only 14% of children aged 6–23 months met the Minimum Dietary Diversity (MDD) criteria [22]. Additionally, the 2016 Ethiopian National Nutritional Supplementation Survey reported that vitamin A Supplementation (VAS) coverage among children was 63%, falling short of the national target of over 90% [23]. The national prevalence of subclinical Vitamin A deficiency (serum retinol < 0.7 μmol/L) was alarmingly high at 37.7% [24,25].

Micronutrient deficiencies are significant public health issues in Ethiopia, primarily due to diets lacking diversity, low bioavailability, frequent meal skipping, limited access to micronutrient-rich and fortified foods, and low intake of vegetables and fruits [26–28]. To combat these deficiencies among children, the national nutritional supplementation program provides both food and supplements. For children older than six months, recommended micronutrients include foods rich in vitamin A and iron, multiple micronutrient powders, iron and vitamin A supplements (VAS), and deworming for those older than 12 months [29–31]. VAS and deworming are administered semi-annually to children aged 6–59 months as part of the national nutrition program. For children aged 6–23 months, it is recommended that they consume four or more food groups rich in iron and vitamin A within the previous 24 hours, receive multiple micronutrient powders and iron supplementation within seven days, and have over 75% deworming coverage within the last six months. Interventions to improve maternal nutrition include multiple micronutrient supplements, food fortification, supplementary food, nutrition education, and counseling, primarily through community-based nutrition programs in Ethiopia [32].

Despite global efforts, progress towards the 2025 World Health Assembly's nutrition targets, particularly in reducing child wasting and stunting through adequate dietary intake, remains insufficient [33]. Meeting the 2030 Sustainable Development Goals (SDGs), especially SDG 2, which aims to end hunger, achieve food security, and improve nutrition, also presents significant challenges [34]. This lack of progress is most pronounced in low- and middle-income countries, including Ethiopia [35]. The Ethiopian government has made notable progress in addressing various forms of malnutrition. To further this goal, it has developed a comprehensive Food and Nutrition Policy (FNP) [36]. Additionally, the Seqota Declaration underscores the government's commitment to eradicating undernutrition by 2030 [37,38]. Although the Ethiopian government has implemented various measures, the fight against malnutrition is advancing slowly, leaving many children to suffer from its severe impacts [39–41].

While there is documented evidence of inadequate micronutrient intake among both agrarian communities and urban dwellers in Ethiopia [42,43], there is limited information on the micronutrient intake of children aged 6–23 months in the Afar. These areas are predominantly inhabited by pastoralist communities with limited cultivation of fruits and vegetables [44]. Furthermore, the Afar regions are recognized as hotspots for high food insecurity, elevated child malnutrition rates, poor infrastructure, inaccessible health and recurrent droughts [45–48]. As far as literature searched, there is a scarcity of studies examining the

individual and community-level minimum recommended micronutrient intake status and factors influencing micronutrient intake among children aged 6–23 months in the study area. Therefore, this study aimed to evaluate the minimum recommended micronutrient intake status and associated factors among pastoralist children aged 6–23 months in the study area.

## Methods

### Study area and period

Afar National Regional State is one of the twelve regional states of Ethiopia. Samara is the capital city of the Afar national regional state, which is 570 km from Addis Ababa and located in the north-eastern part of the country. The size of Afar National Regional State is 278,000 sq. km. Geographically, the region is located between 9°N and 12°N latitude and 40°E and 42°E longitude at the northern tip of the Great East African Rift Valley and is found at 432 m altitude. Afar regional state has six administrative zones, such as Awsi Rasu (zone one), Kilbat Rasu (zone two), Gabi Rasu (zone three), Fanti Rasu (zone four), Hari Rasu (zone five), and Yagudae Rasu (zone six).

Based on the 2007 Census conducted by the Central Statistical Agency of Ethiopia, the region has a population of 1,390,273, consisting of 775,117 men and 615,156 women, and a total population projection for 2022 was 2,090,910, consisting of 1,137,519 men and 953,391 women [49]. The Afar community is predominantly pastoralist; over 90% depend on extensive livestock production for survival, as there are virtually no alternative forms of employment [50]. Their nomadic lifestyle creates major obstacles to accessing essential social services. Consequently, children are prone to malnutrition and communicable diseases, and difficulty and expensive to reach health services. This systemic marginalization is exacerbated by their exclusion from the economic and political decisions that impact their lives [50].

For this study, Aysaita district was selected randomly, which was found in Awsi Rasu (Administrative Zone 1) within the Afar Region, Ethiopia. It is located 600 km northeast of Addis Ababa and 72 km away from Samara town. The district is administratively divided into 11 kebeles. According to the Aysaita District health bureau, the total human population of the district is 61961 [51]. Among the total population of the district, 12392 households have children aged 6–23 months [51]. The study was conducted from August 1–30, 2024.

### Study design

A community-based cross-sectional study was conducted

### Population

**Source population.** All children aged 6–23 months residing in the Aysaita district were source populations.

**Study population.** All randomly selected children, aged 6–23 months, who were residents of the selected kebeles in the Assayta district were study populations.

### Eligibility criteria

**Inclusion criteria.** The study included all breastfed and non-breastfed children aged 6–23 months who had been living in the designated study area for at least 6 months before the survey and were residing with their biological mothers aged 18–49 years.

**Exclusion criteria.** Children have severe acute malnutrition or chronic illnesses requiring specialized dietary management were excluded from the analysis. Mothers who have a health problem that can affect the interview process also excluded.

### Sample size determination

The sample size was determined by ***OpenEpi online software version 3.1*** using the single population proportion formula by considering the estimated proportion of adequate micronutrient intake status, 59.1% in Ethiopia, from the study

conducted in sub-Saharan Africa in 2023 [52], with a 95% confidence level and a 5% marginal error. As a result, **372** samples were obtained. Since it is a multistage sampling technique, using the design effect multiplied by 1.5 becomes **558**, and by adding a 10% non-response rate, the final sample size becomes **614.** The sample size determination for factors associated with the minimum recommended micronutrient intake status was calculated also using *OpenEpi online software version 3.1.* The calculation was based on key determinants identified from previous literature: maternal education, antenatal care attendance, and child age (13–23 months) [53,54] (Table 1). For each factors the following assumptions were applied: a 95% confidence level (two-sided), 80% statistical power, and a 1:1 ratio of unexposed to exposed, percent of exposed and unexposed with outcome variable and odd ratio. However, considering larger sample size, a sample size of **614**, calculated using the single population proportion formula for the primary objective, was adopted for the study.

## Sampling procedure

Multi-stage sampling technique was used to select the study participants. First from eleven kebeles of the district, five kebeles selected by lottery methods. The sample size allocated to each selected kebele's by proportional to the number of children aged 6–23 months (Fig 1). The value of 'K' was calculated from N/n; where N = total population of selected kebeles, n = sample size (K = N/n = 5803/614 = 9). Accordingly, every 9th child participated in the study from each kebeles. The expected total number of participants in the selected kebeles were n = 614. Then, the study participant was selected by systematic sampling with skip interval of 9 at each kebele.

The sample frame of households (mothers who deliver in the last two years and have children aged 6–23 months) was prepared in all selected kebeles from the mother's registration book, which was found from health extension workers. In addition, the survey was conducted to identify a list of the household (mothers who delivered in the last two years and have children aged 6–23 months) in case the list of the household was not registered in the kebeles before actual data collection.

**Table 1. Calculated sample size for factors associated with minimum recommended micronutrient intake status using two populations proportions by *OpenEpi online software version 3.1.*, 2024.**

| No | Variables | Ratio | CI | Power | % exposed | % unexposed | AOR | Sample (10% non-response and 1.5 design effect |
|---|---|---|---|---|---|---|---|---|
| 1. | Antenatal care visit | 1:1 | 95% | 80% | 73.6 | 26.3 | 1.9 | 70 |
| 2. | Child age (13–23 months) | 1:1 | 95% | 80% | 68.4 | 31.6 | 1.7 | 109 |
| 3. | Educated mother | 1:1 | 95% | 80% | 41.7 | 58.4 | 2.09 | 502 |

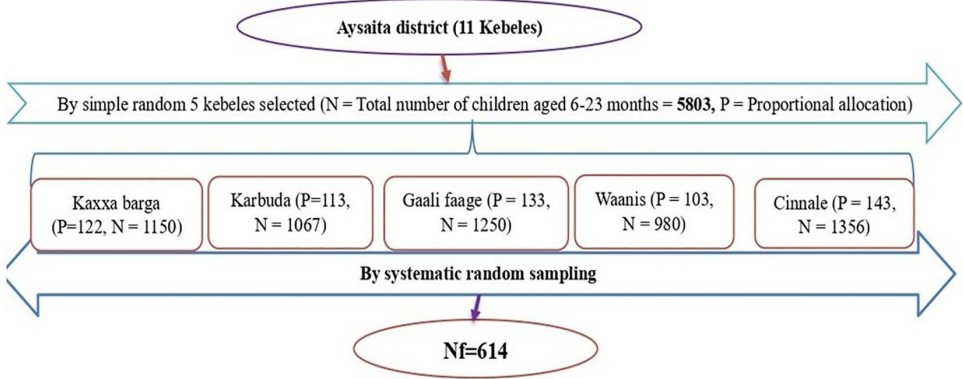

**Fig 1. Schematic presentation of the sampling procedure of study participants in assessment of minimum recommended micronutrient intake status and associated factors among children aged 6-23 months in Aysaita district, Afar region, Ethiopia, 2024.**

For mothers with twins, only one was selected by lottery methods. Subsequently, every 9th child aged 6–23 month was included until the desired sample size achieved. In households in which candidate respondents were not at home, but it was known that there were eligible households for the study, the interviewers were revisited the household at three different time intervals, and as a result no any interviewers fail to meet the household, and the household was not replaced by the next household in a clockwise direction.

## Study variable

**Dependent variable.** Minimum recommended micronutrient intake status

**Independent variable. Socio-demographic and economic characteristics:** Age of mothers, sex of the child, place of the residence, family size, ethnicity, religion, marital status, educational status, husband educational status, Occupational status, husband educational status, family monthly income (USD), functional electricity, functional radio, functional television, functional refrigerator, and bank account.

**Obstetric characteristics of the mothers:** Age at first pregnancy, Antenatal care follow up, desire for more childrens, place of delivery, mode of delivery, postnatal check up, had exclusive breast feeding, current pregnancy status, and media exposure.

**Child characteristics and common childhood illness:** Child age in months, had diarrheas in the past two weeks, had cough in the past two weeks, and had fever in the past two weeks,

## Operational definitions and variable measurements

**Minimum recommended micronutrient intake status:** It's the micronutrient nutrient (MN) intake status among children aged 6–23 months, which was determined by respondents' reports and assessment of intake status. So, there were six options: food rich in vitamin A (VA) or food rich in iron in the last 24 hours, multiple micronutrient powder (MNP) in the past seven days or iron supplement (iron pills, sprinkles with iron, or iron syrup) consumed within the previous seven days, vitamin A supplementation (VAS) or deworming within the previous six months [29–31,55]. Accordingly, if the respondent reported that the child had eaten' or consumed at least one of these minimum recommended micronutrient nutrients (MNs), its considered as "Yes"; if the children received none of these minimum recommended MNs, it was considered as "No" [56].

**Vitamin A-rich foods**: were assessed based on the consumption of seven food groups within the past 24 hours. These groups included: 1) Eggs, 2) Meat (beef, lamb, chicken), 3) Pumpkin, carrots, and squash, 4) Dark green leafy vegetables, 5) Mangoes, papayas, and other Vitamin A-rich fruits, 6) Liver, heart, and other organs, and 7) Fish or shellfish. If the respondent indicated that the child had consumed at least one item from these groups, it was recorded as a 'yes' for Vitamin A-rich food; otherwise, it was recorded as a 'no' [29–31,55].

**Iron-rich foods**: were evaluated based on the consumption of four specific food groups within the last 24 hours. These groups included: 1) Eggs, 2) Meat (beef, pork, lamb, and chicken), 3) Liver, heart, and other organs, and 4) Fish or shellfish. If the respondent indicated that the child had consumed at least one item from these groups, it was recorded as a 'yes' for iron-rich food; otherwise, it was recorded as a 'no' [29–31,55].

**Multiple micronutrient powder:** The assessment of multiple micronutrient powders was conducted by asking respondents if their child had received any micronutrient powders in the past seven days [29–31,55].

**Iron supplementation:** was evaluated by asking respondents if their child had received iron supplements, such as iron pills, sprinkles with iron, or iron syrup, within the past seven days [29–31,55].

**Vitamin A supplementation (VAS) and deworming** for children aged 6–23 months were evaluated by checking if they had received these treatments in the past six months. This assessment was done by reviewing the integrated child health card, which includes immunization and growth monitoring records, as well as obtaining verbal confirmation from the mother [29–31,55].

**Media exposure** was assessed as "yes" if they have access to all three media (newsletter, radio, and television) at least once a week, otherwise "no" if they did not have any media exposure [56].

## Data collection tools and technique

The questionnaire was prepared in the English language and the tools for data collection are developed from review of guidelines and related literatures (S1 Questionaries) [29–31,55,56].The contents of the questionnaire included the following four sections: The first section assessed socio-demographic and economic characteristics. The second part deals with obstetric characteristics of the mothers, the third section assessed about child characteristics and common childhood illness and fourth part assessed minimum recommended micronutrient intake which holds 11 questions [29–31,55,56]. The mothers of the selected children were interviewed, and the data were collected using the Kobo Toolbox, an electronic data collection tool.

A pretested structured questionnaire was used through a face-to-face interview to gather data from mothers. A team of four skilled data collectors, supported by one expert supervisor with backgrounds in public health nutrition, conducted and supervised the data collection process.

## Data quality assurance and control

To ensure quality of data, the questionnaire was translated into the local language (Afar af and Amharic language) by nutrition experts. Finally, before data collection, it was retranslated back to English to verify consistency. Before starting the actual data collection, one day of training was given for the data collectors and supervisors. A pre-test was conducted on 21 (5%) of the total sample size at Gala.ilu Kebele, which was not selected for study by data collectors, and since internal consistency was adequate, no amendments were made. The reliability of the questionnaires was checked through STATA by reliability index measurement for minimum recommended micronutrient status questions, which showed adequate internal consistency (Cronbach's alpha) of **0.9014.**

The principal investigator and supervisors conducted day-to-day on-site supervision during the whole period of data collection. At the end of each day, the questionnaires were reviewed and checked for completeness and accuracy by the supervisors and investigators. Then corrective modification was made by the principal investigators. Data were checked for completeness, accuracy, clarity, and consistency before being entered into the software. Proper coding and categorization of data was maintained for the quality of the data to be analyzed.

## Data processing, analysis and presentation

The collected data (S1 Dataset) was exported to STATA version 17 for further statistical analysis. Data exploration, editing, and cleaning were undertaken before analysis. Descriptive statistics such as median and interquatile range values for continuous data; percentage, frequency, and tables for categorical data were used. A binary logistic regression analysis was conducted to see factors associated with the minimum recommended micronutrient status and select associated variables with a 95% confidence interval and a P value less than 0.05 for multivariable logistic regression to control the effect of confounders. Then, multivariable analysis was performed to determine the independent predictors of the dependent variable.

Regarding multi-collinearity among independent variables, STATA itself was controlled and checked via variance inflation factor (VIF) and tolerance. Variable VIF greater than 10 and tolerance less than 0.1 were removed in STATA. Accordingly, no multi-collinearity was observed. Final model fitness was checked by the Hosmer and Lemeshow tests, and model adequacy was declared when p-value > 0.05. Accordingly, the final model of Hosmer and Lemeshow showed **0.1149**. The significance was checked and declared using a p-value less than 0.05 and a 95% confidence interval in the final model. Strength of association was interpreted by using an adjusted odds ratio with a 95% confidence interval.

## Ethical statements

Ethical approval was obtained from the *Institutional Ethical Review Committee (IERC) of Samara University, College of Medicine and Health Science, with reference number (Ref. No.: SU/CMHS/IERC/814/2016)*. Following the approval by IERC, official letters of cooperation were written to the Aysaita district health administration office, and in turn, the woreda health administration office wrote letters to each selected kebeles and village's administration office in order to get permission and cooperation.

Prior to participation, trained data collectors obtained *verbal informed consent* from the mothers or caregivers of all participants after providing a comprehensive, locally-translated explanation of the study's purpose, contents, methodology, anticipated benefits, and potential risks. The *Institutional Ethical Review Committee (IERC)* granted a waiver for written documentation of consent, as the research presented minimal risk and study population was predominantly from a rural community with high rates of illiteracy. For each participant, the data collector signed a standardized consent log to confirm that the study had been explained, questions were answered, and agreement was provided. To enhance community trust and procedural rigor the process was witnessed by local community health worker for a subset of the participants. To ensure confidentiality, the names of respondents were replaced by code numbers. Furthermore, participants were given the chance to ask any doubt about the study and made free to refuse or stop the interview at any moment they wanted.

## Results

### Sociodemographic and socioeconomic characteristics

In this study, all 614 mothers/caregivers with children aged 6–23 months participated, giving a response rate of 100%. The mean age of the mothers was 33.03 (SD±4.99). Of these, 314(51.14%) were found in the age group of 35–49 years. The majority of the participating children were female, 352 (57.33%). Almost all of the respondents, 614(100%), were followers of the Muslim religion, and 574 (93.49%) of the study participants belong to the Afar ethnicity. In this study, 556 (90.55%) of the mothers were married and 483 (78.66%) were housewives.

Among included mothers, 532 (86.64%) were unable to read and write, followed by read and write education level, which represented 70 (11.40%). The mean family size of the household was 6.4 (SD±3.75). Of these, most of the study participants, 378 (61.56%) have six and above family size. The mean monthly income of the household was 9.68 USD (SD±8.28). Of these, most of the household, 544(88.6%) had monthly income less than or equal to 7.07 USD. Media exposure was absent in 554 households, representing 90.52% of the total surveyed (Table 2).

### Obstetric characteristics of the mothers/care givers

The mean age of the mother's first pregnancy was 15.56 (SD±2.94). Of these, the majority of them, 521 (84.85%) belonged to the 12–18 age range. In terms of their desire for more children, 566 (92.18%) mothers desire to have more children, while 41 (6.68%) desire no more children over time. Three hundred and fourty-four (56.03%) of the mothers had delivered at home, while 270 (43.97%) of the mothers had delivered in the health institution. About 569 (92.67%) mothers had delivered spontaneous vaginal, while 45 (7.33%) had delivered by cesarean section. Out of the participating mothers, 332 (54.07%) of women had PNC checkups, and 282 (45.93%) stated that they were not pregnant or unsure about their pregnancy status (Table 3).

### Child characteristics and common childhood illness

Participated children had a median age of 8 months with an interquartile range (IQR=7–12 months). Among the included children, about one hundred twenty-four (20.20%) had exclusive breast-feeding status. Out of the participating children, 344 (54.40%) had diarrhea in the past two weeks, and 301 (49.02%) of them had cough in the past two weeks. Regarding fever, 252(41.04%) of children had fever in the past two weeks (Table 4).

**Table 2. Socio demographic and economic characteristics of children age 6-23 months and mother/caretakers at Aysaita district, Afar Region, Ethiopia, 2024.**

| Variables | Category | Frequency | Percentage (%) |
|---|---|---|---|
| Age of mother/caregiver | 15-24 | 41 | 6.68 |
| | 25-34 | 259 | 42.18 |
| | 35-49 | 314 | 51.14 |
| Sex of the child | Male | 262 | 42.63 |
| | Female | 352 | 57.33 |
| Residence | Urban | 46 | 7.49 |
| | Rural | 568 | 92.51 |
| Religion of mother | Muslim | 614 | 100 |
| Ethnicity | Afar | 574 | 93.49 |
| | Amhara | 40 | 6.51 |
| Marital status | Married | 556 | 90.55 |
| | Separated | 40 | 6.51 |
| | Widowed | 18 | 2.93 |
| Occupation | Gov't employed | 27 | 4.40 |
| | Housewife | 483 | 78.66 |
| | Merchant | 12 | 1.95 |
| | Pastoralist | 92 | 14.98 |
| Occupation of the husband/partners | Daily worker | 17 | 2.77 |
| | Merchant | 19 | 3.09 |
| | Pastoralist | 490 | 79.80 |
| | Farmer | 66 | 10.75 |
| | Gov't employed | 22 | 3.58 |
| Education status | Unable to read and write | 532 | 86.64 |
| | Read and write | 70 | 11.40 |
| | Primary education | 5 | 0.81 |
| | Secondary education | 7 | 1.14 |
| Educational status of the husband/ partners | Unable to read and write | 483 | 78.66 |
| | Read and write | 97 | 15.8 |
| | Primary education | 5 | 0.81 |
| | Secondary education | 17 | 2.77 |
| | College and above | 12 | 1.95 |
| Family size | < 6 | 236 | 38.44 |
| | ≥ 6 | 378 | 61.56 |
| Monthly income in USD | ≤ 7.07 USD | 544 | 88.6 |
| | 7.07-35.34 USD | 53 | 8.63 |
| | ≥ 35.34 USD | 17 | 2.77 |
| Media exposure | Yes | 58 | 9.48 |
| | No | 554 | 90.52 |

**Note: USD**: United State Dollar.

## Minimum micronutrient intake status of the childrens aged 6–23 months

A micronutrient intake composite score was constructed from a series of six closed-ended questions to evaluate the minimum recommended micronutrient intake status in children aged 6–23 months. Data on consumption of vitamin A (VA) or

**Table 3. Obstetric characteristics of the mothers/care givers of children aged 6–23 months at Aysaita district, Afar Region, Ethiopia, 2024.**

| Variables | Category | Frequency | Percentage (%) |
|---|---|---|---|
| Age at 1st pregnancy | 12-18 | 521 | 84.85 |
| | ≥ 19 | 93 | 15.15 |
| Antenatal care follow-up | Yes | 386 | 62.87 |
| | No | 218 | 35.5 |
| | I don't know | 10 | 1.63 |
| Desire for more children | Undecided | 7 | 1.14 |
| | Wants | 566 | 92.18 |
| | Wants no more | 41 | 6.68 |
| Place of delivery | Health facility | 270 | 43.97 |
| | Home | 344 | 56.03 |
| Mode of delivery | Cesarean section | 45 | 7.33 |
| | Spontaneous vaginal delivery | 569 | 92.67 |
| Post natal check up | Yes | 332 | 54.07 |
| | No | 282 | 45.97 |
| Current pregnancy status | Non pregnant or unsure | 401 | 65.31 |
| | Pregnant | 213 | 34.69 |

**Table 4. Child characteristics and common childhood illness of children aged 6–23 months at at Aysaita district, Afar Region, Ethiopia, 2024.**

| Variables | Category | Frequency | Percentage (%) |
|---|---|---|---|
| Current age of the child (months) | 6-11 months | 413 | 67.26 |
| | 12-23 months | 201 | 32.74 |
| Exclusive breastfeeding status | Yes | 124 | 20.20 |
| | No | 490 | 79.80 |
| Had diarrhoea in the past two weeks | Yes | 334 | 54.40 |
| | No | 280 | 45.60 |
| Had cough in the past two weeks | Yes | 301 | 49.02 |
| | No | 313 | 50.98 |
| Had fever in the past two weeks | Yes | 252 | 41.04 |
| | No | 362 | 58.96 |

iron-rich foods within the last 24 hours, multiple micronutrient powder (MNP) or iron supplements (pills, sprinkles, or syrup) within the past seven days, and receipt of vitamin A supplementation (VAS) or deworming within the previous six months were collected via survey.

Responses to these dichotomous questions were coded using a binary scoring system: a value of "1" was assigned for receiving a micronutrient intervention or consuming a relevant food, and a value of "0" was assigned for non-receipt or non-consumption. The composite variable was then created to categorize each child's overall micronutrient status. A child was classified as "Yes" or "Received" if they scored positively (i.e., received a score of 1) on at least one of the six indicators. Conversely, a child was categorized as "No" or "Not Received" if they did not receive any of the assessed micronutrient interventions.

Accordingly, this study showed that, 219 (35.67%) children between the ages of 6 and 23 months had received at least one minimum recommended micronutrient intake status (35.67%; 95% CI: 31.96, 39.54%) (Fig 2). Overall, 64.33% (95% CI: 60.45, 68.03) of children between 6 and 23 months old hadn't received any MNs from the recommended sources. Fourten

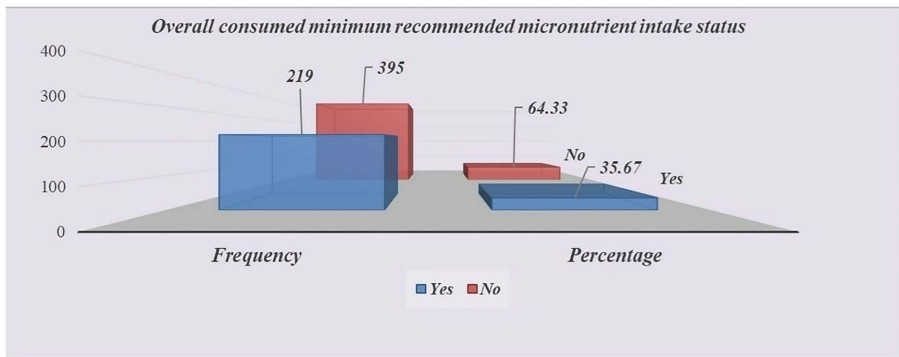

**Fig 2. Minimum recommended micronutrient intake status among children aged 6–23 months at Aysaita district, Afar Region, Ethiopia, 2024.**

point eight two (14.82; 95% CI: 12.21–17.86) of the children received iron supplements, 13.84% (95% CI: 11.32–16.81) of the children received multiple micronutritn powder (MNP) within the last seven days, and 23.45% (95% CI: 20.26–26.97) of the children received vitamin A supplementation (VAS) within the previous six months. About 24.59% (95% CI: 21.34–28.16) of the children received deworming medication within the last six months. Merely 28.99% (95% CI: 25.53–32.71) of the children had meals rich in vitamin A (VA), and 22.96% (95% CI: 19.80–26.46) had food rich in iron in the previous 24 hours.

### Factors associated with minimum recommended micronutrient intake

Based on bivariate logistic regression, being a male child, media exposure, having a postnatal checkup, having exclusive breast feeding, spontaneous mode of delivery, and income category 7.07–35.34 USD were significantly associated with the minimum recommended micronutrient intake status, and all were transferred to multivariable logistic regression analysis to control the effect of confounders. However, in the multivariable logistic regression analysis, being a male child, media exposure, having exclusive breast feeding, spontaneous vaginal delivery, and income category 7.07–35.34 USD were identified as predictors of minimum recommended micronutrient intake status (p < 0.05) (Table 5).

**Table 5. Bivariable and multivariable analysis of factors associated with minimum recommended micronutrient intake status among children aged 6–23 months at Aysaita district, Afar Region, Ethiopia, 2024.**

| Variables | Category | Yes | No | COR (95%CI) | AOR (95%CI) | P-value for AOR |
|---|---|---|---|---|---|---|
| Sex of the child | Male | 118(45.04) | 144(54.96) | *2.03(1.45, 2.84) * * | *2.04(1.42, 2.92) ** * | *0.000** * |
| | Female | 101(28.69) | 251(71.31) | 1 | 1 | |
| Media exposure | Yes | 35(60.34) | 23(39.66) | *3.11(1.78, 5.41) * * | *2.50(1.34, 4.64) ** * | *0.004** * |
| | No | 182(32.85) | 372(67.15) | 1 | 1 | |
| Post natal check up | Yes | 144(43.37) | 188(56.63) | *2.11(1.50, 2.97) * * | 1.20(0.81, 1.77) | 0.94 |
| | No | 204(41.60) | 286(58.37) | 1 | 1 | |
| Exclusive breast feeding | Yes | 204(41.63) | 286(58.37) | *5.18(2.93, 9.15) * * | *5.09(2.72, 9.52) ** * | *0.000** * |
| | No | 15(12.10) | 109(87.90) | 1 | 1 | |
| Mode of delivery | Spontaneous | 209(36.73) | 360(63.27) | *2.03(1.01, 4.18) * * | *2.71(1.20, 6.10) ** * | *0.016** * |
| | Cesarian section | 10(22.22) | 35(77.78) | 1 | 1 | |
| Income category | ≤ 7.07 USD | 189(34.74) | 355(65.26) | 1 | 1 | |
| | 7.07-35.34 USD | 30(42.86) | 40(57.14) | *2.10(1.19, 3.71) * * | *2.90(1.49, 5.65) ** * | *0.002** * |

*Note: **Chi-square test** used for categorical variables, * Shows association in bivariable analysis, ** for association in multivariable analysis, USD = United State Dollar, **COR** = Crude Odd Ratio, **AOR** = Adjusted Odd Ratio, **A p-value** of < 0.05 was considered statistically significant in the final model.*

Accordingly, findings from multivariable logistic regression showed that the odds of receiving minimum recommended micronutrient intake were 2.71 times higher for children whose mothers gave birth by spontaneous vaginal than for children whose mothers gave birth by cesarian section (AOR: 2.71, 95% CI: 1.20, 6.10). The odds of receiving minimum recommended micronutrient intake among children whose mothers had been exposed to media were 2.50 more likely as compared to those not exposed to media (AOR = 2.50; 95% CI: 1.34, 4.64). Being a male child was 2.04 times [AOR = 2.04; 95% CI: 1.42, 2.92] more likely to receive the minimum recommended micronutrient intake as compared to those of females. Additionally, the odds of receiving minimum recommended micronutrient intake were 5.09 times higher for those children who had exclusive breast feeding than those who had no exclusive breast feeding (AOR: 5.09, 95% CI: 2.72, 9.52). Furthermore, households that had a monthly income category of 7.07–35.34 USD were 2.90 more likely to receive the minimum recommended micronutrient intake than those who had no monthly income category (AOR: 2.90, 95% CI: 1.49, 5.65).

## Discussion

The Ethiopian national guidelines for the prevention and control of micronutrient deficiency state that micronutrients must be given to all children [57]. Our study, however, showed that only 35.67% (95% CI: 31.96, 39.54%) of 6–23-month-old'children had received any of the minimum recommended MN sources in the study area. Based on the findings, the minimum recommended micronutrient intake status in the study area showed there was a gap on implementing and meeting the National Food and Nutrition Strategy (NFNS-2021) targets [58], indicating that a significant percentage of its 6- to 23-month-old children do not consume the recommended number of micronutrients as per national guidelines and nutritional strategy.

This finding is comparatively higher than studies conducted in Ethiopia, 27.6% [53]. However, the finding is comparatively lower than to previous studies conducted in Ethiopia 62.7% [56], sub-Saharan Africa 73.99% [52], Rwanda (68%), Burundi (78%) [59], and India (58.1%) [60]. The possible discrepancy for this prevalence may be justified by the fact that variation in dietary intake across regions due to cultural, socioeconomic disparities, rural and marginalized communties, nutritional taboos, and limited availability of micronutrient food could be a probable explanation [56]. Additionally, each geographic region has a different staple food due to differences in literacy, food insecurity, traditions, and beliefs [61].

After adjusting the odd ratio in the final multivariable logistic regression, being a male child, media exposure, having exclusive breast feeding, spontaneous vaginal delivery, and income category 7.07–35.34 USD were significantly associated factors with the minimum recommended micronutrient intake status among children aged 6–23 months.

Children whose mothers were exposed to media were more likely to consume minimum recommended micronutrients compared to those whose mothers were not. This finding aligns with studies conducted in Ethiopia [42,62,63], Nigeria [64], Indonesia [65], and Nepal [66]. The influence of national radio and television promotions on child nutrition-related advertisements likely contributed to this outcome [62]. Additionally, media can influence behavioural change by influencing mother attitude and practices, promoting healthier food choices, encouraging new adoption and benefial habits as well as socioeconomic improvements [67–70].

This study found that children whose mothers gave delivery by spontaneous vaginal delivery had higher odds of minimum recommended micronutrient intake compared to those whose mothers gave delivery by cesarian section. One possible explanation is that mothers who gave birth by cesarean section associated with lower rate of early breastfeeding initiation and shorter duration of exclusive breast feeding compared to vaginal delivery [71,72]. Additionally, C-section born infants may have imbalances in immune cell populations, which can put them at a higher risk for certain immune disorder later in life which may affect overall metabolic and absorptive capabilities [72].

This study also found that children who had exclusive breast feeding had higher odds of minimum recommended micronutrient intake compared to those whose mothers did not. This might be due to the fact that breast milk is nutritionally superior to infant formula due to its optimal, easily digestible composition and the high bioavailability of its nutrients,

meaning minerals like iron are far more absorbable [73]. Its micronutrient content remains reliably stable, contingent on a balanced maternal diet. Furthermore, exclusive breastfeeding protects infants from infections like diarrhea, which helps prevent nutrient malabsorption and ensures they can effectively use the nutrients they consume [73].

Moreover, this study found that being a male child had higher odds of minimum recommended micronutrient intake as compared to those of females. This finding is similar to those of a study conducted in Ethiopia in Diredawa city [74] and Haramaya town [75]. This might be due to the fact that deep seated cultural norms that favors sons as future bread-winners, a gender bias in nutrition exists, leading to preferential allocation of food and healthcare to male children over females [76,77]. This intra-household inequality is more pronounced in resource poor setting and mitigated by higher levels of maternal education and empowerement [78].

Furthermore, this study revealed that children whose households had higher monthly income were more likely to intake the minimum recommended micronutrient intake than households that had a monthly income less than 7.07 USD. This finding is consistent with a study conducted in Ethiopia and a secondary data analysis of the Nepalese Demographic and Health Survey [66]. Similarly, the 2011 Ethiopian DHS reported that children from the wealthiest families were more likely to consume four or more food groups [79]. This could be because higher household income directly improves child nutrition by enabling families to afford a wider variety nutritios like fruits, vegetables, and animal source products rich in vital micronutrient [80].

## Conclusions

In conclusion, the study reveals that despite national guidelines advocating for micronutrient supplementation for all children, only 35.67% of children aged 6–23 months in the study area received the minimum recommended micronutrients. The results suggest that the implementation of the National Food and Nutrition Strategy (NFNS-2021) is falling short of its targets. Specifically, a large proportion of children aged 6–23 months are failing to consume the minimum level of micronutrients prescribed by national guidelines and the nutritional strategy. Being a male child, media exposure, exclusive breastfeeding, spontaneous vaginal delivery, and household income were significantly associated with the minimum recommended micronutrient intake for children aged 6–23 months.

District health bureaus should leverage media campaigns, including national radio and television, to expand and intensify public awareness and education on child micronutrient intake. Additionally, health initiatives must implement targeted educational programs to address cultural biases that favor male children, as evidence suggests gender-based disparities in feeding practices, which result in male children being more likely to meet nutritional intake recommendations. Therefore, ensuring equal nutritional attention for female children is a critical component of improving overall child health outcomes.

Advocacy by health professionals is crucial for promoting exclusive breastfeeding (EBF) for the first six months of life, as practice linked to optimal infant growth and enhanced micronutrient intake. Encouraging spontaneous vaginal deliveries where medically feasible is recommended, as it is associated with fewer maternal and infant complications and facilitates the earlier initiation of breastfeeding compared to cesarean sections. Additionally, interventions should involve non-governmental organizations (NGOs) providing financial support and income-generating resources to improve diet diversity and nutritional security for low-income families. Furthermore, future research should conduct longitudinal studies to definitively establish the causal relationship between inadequate micronutrient intake and its associated factors as well as qualitative study to detail investigate barriers to minimum micronutrient intake status.

## Supporting information

**S1 Questionaries. English version of the structured questionnaire used for data collection.** The questionnaire contains sections on socio-demographic and economic characteristics, obstetric characteristics of the mothers, child characteristics and common childhood illness and minimum recommended micronutrient intake which holds 11 questions. (PDF)

**S1 Dataset. Identified minimum recommended micronutrient intake dataset underlying results and findings in this study.** The datasets include variables related with sociodemographic, obstetrics characteristics, child character tics and common childhood illness and variable of interest (minimum recommended micronutrient intake status). (ZIP)

## Acknowledgments

The authors would like to thank Samara University, College of Medicine and Health Science, regional health office, zone health office, woreda health bureau and administrative office, Woreda health extension worker, data collectors, supervisors, and study participants for their cooperation in the study.

## Author contributions

**Conceptualization:** Abdulkerim Hassen Moloro.

**Formal analysis:** Ali Mohammed Ibrahim, Abdulkerim Hassen Moloro, Yeshimebet Ali Dawed.

**Investigation:** Abdulkerim Hassen Moloro.

**Methodology:** Ali Mohammed Ibrahim, Abdulkerim Hassen Moloro, Dereje Desalegn Boshe, Yeshimebet Ali Dawed.

**Software:** Ali Mohammed Ibrahim, Abdulkerim Hassen Moloro, Dereje Desalegn Boshe, Yeshimebet Ali Dawed.

**Supervision:** Yeshimebet Ali Dawed.

**Validation:** Abdulkerim Hassen Moloro.

**Writing – original draft:** Abdulkerim Hassen Moloro.

**Writing – review & editing:** Ali Mohammed Ibrahim, Abdulkerim Hassen Moloro, Dereje Desalegn Boshe, Yeshimebet Ali Dawed.

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
