## [Decision Letter · Decision Letter 0]

25 Mar 2025

Dear Dr. Moloro,

Thank you for submitting your manuscript to PLOS ONE. After careful consideration, we feel that it has merit but does not fully meet PLOS ONE’s publication criteria as it currently stands. Therefore, we invite you to submit a revised version of the manuscript that addresses the points raised during the review process.

We look forward to receiving your revised manuscript.

Kind regards,

Dinaol Abdissa Fufa, Mph

Academic Editor

PLOS ONE

2. In the ethics statement in the Methods, you have specified that verbal consent was obtained. Please provide additional details regarding how this consent was documented and witnessed, and state whether this was approved by the IRB.

3. In the online submission form, you indicated that “All data are available based on the reasonable request from principal investigator or corresponding author”

5. Please upload a copy of Figure 3, to which you refer in your text on page 23. If the figure is no longer to be included as part of the submission please remove all reference to it within the text.

Reviewers' comments:

Reviewer's Responses to Questions

**Comments to the Author**

1. Is the manuscript technically sound, and do the data support the conclusions?

Reviewer #1: Yes

Reviewer #2: Yes

Reviewer #3: Yes

Reviewer #4: Yes

2. Has the statistical analysis been performed appropriately and rigorously?

Reviewer #1: Yes

Reviewer #2: Yes

Reviewer #3: Yes

Reviewer #4: Yes

3. Have the authors made all data underlying the findings in their manuscript fully available?

Reviewer #1: Yes

Reviewer #2: Yes

Reviewer #3: Yes

Reviewer #4: Yes

4. Is the manuscript presented in an intelligible fashion and written in standard English?

Reviewer #1: Yes

Reviewer #2: Yes

Reviewer #3: Yes

Reviewer #4: Yes

Reviewer #1: Dear authors, thank you so much for your effeort in this research, really it's an excellent work and I enjoyed reading all parts of it. I only have few recommendations

- It will be better to add a figure representaing Minimum recommended micronutrient intake status among children either vtotal or for element. I think it It will explain the results better

-Few paraphrasing and gramatical revision is needed

Reviewer #2: Manuscript is well-written with profound and detailed information regarding the topic. Methodology is well explained and the sampling method is written in proper sequential order. Please add legends under the table, mentioning the statistical tests used and also value at which they are considered significant.

Reviewer #3: the methodology part is very clear. the result part , particularly the variable of your interest, the way you calculated is not clear, if possible make it clear. the scientific explanation of the discussion if possible support with references. remove conclusion and recommendation from the discussion part. re-write the conclusion based on your key findings. The recommendation part should be remove and substitute very relevant that should be goes with your conclusion.

Reviewer #4: This study is well-conducted, and the researchers are commended for their work thorough the title of Minimum Recommended Micronutrient Intake Status and Associated Factors among Children Aged 6-23 Months in Pastoralist Community of Afar Region, Ethiopia, 2024 : Community Based Cross-sectional Study. The study is effectively structured, with a clear identification of the impact of Minimum Recommended Micronutrient Intake Status with well thought recommendations for public health interventions. However, to enhance its credibility and ensure clear communication of the findings, the researchers should address the minor comments and suggestions provided above. Once these revisions are made, the study will be ready for publication and will provide valuable insights for controlling Micronutrient deficiency in your study area and the country as well.

**Do you want your identity to be public for this peer review?** For information about this choice, including consent withdrawal, please see our Privacy Policy

Reviewer #1: No

Reviewer #2: **Yes: ** Tabeer Tanwir Awan

Reviewer #3: No

Reviewer #4: **Yes: ** Published Name: Ayele HM

First Name: Habtamu Molla

Last Name: Ayele

---

## [Author Response · Author response to Decision Letter 1]

16 Sep 2025

Author response to reviewer and editor

Manuscript ID: PONE-D-24-55242

Manuscript Title: Minimum Recommended Micronutrient Intake Status and Associated Factors among Children Aged 6-23 Months in Pastoralist Community of Afar Region, Ethiopia, 2024: Community Based Cross-sectional Study

1. Response to Reviewer 1

Reviewer 1: Comments

Author Response Page number in the manuscript

1. Dear authors, thank you so much for your effort in this research, really, it's an excellent work and I enjoyed reading all parts of it. I only have few recommendations. Thank you very much for your encouragement and constructive comments.

2. It will be better to add a figure representing Minimum recommended micronutrient intake status among children either total or for element. I think it will explain the results better Revised as commented. We presented using total. 17

3. Few paraphrasing and grammatical revision is needed Revised as commented All pages

2. Response to Reviewer 2

Reviewer 2: Comments

Author Response Page number in the manuscript

1. Manuscript is well-written with profound and detailed information regarding the topic. Methodology is well explained and the sampling method is written in proper sequential order. Thank you very much for your encouragement and constructive comments.

2. Please add legends under the table, mentioning the statistical tests used and also value at which they are considered significant. Revised as commented. We presented using total. 13-18

3. Few paraphrasing and grammatical revision is needed Revised as commented All pages

3. Response to Reviewer 3

Reviewer 3: Comments

Author Response Page number in the manuscript

1. The methodology part is very clear. Thank you very much for your encouragement and constructive comments.

2. The result part, particularly the variable of your interest, the way you calculated is not clear, if possible, make it clear. Now revised and explained detail in result part. 16-17

3. The scientific explanation of the discussion, if possible, support with references. Revised as commented 18-20

4. Remove conclusion and recommendation from the discussion part. re-write the conclusion based on your key findings. The recommendation part should be removed and substitute very relevant that should be goes with your conclusion. Now revised as commented 20-21

4. Response to Reviewer 4

Reviewer 4: Comments

Author Response Page number in the manuscript Remark

1. Firstly, thank you very much for providing the opportunity to review this manuscript. Second, I will list out all the important comments to be incorporated within the manuscript for a better research output. First of All, Thank you for your appreciation and constructive comments.

2. However, the article has no line number for providing comments easily for authors. Therefore, the authors need to use standard page numbering of articles and line number while resubmitting the article which would be important for the second round of comments. Revised as commented

Authors orders

1. Order the Authors in terms of Authors contribution. Corrected as commented. The orders made in the authors contributions subheading 23

ABSTRACT

1. Page 11: line 19. Write the currency of the income category 500-5000 whether it was USD or ETB? Now corrected as commented. Its ETB 3-4

2. Page 11: line 22-23. The Authors need to rewrite all the conclusion and recommendations. 3.1 Conclude and recommend based on your research findings and your study groups (6-23 months). Revised as commented 3-4

3. The study revealed over one-third of the children received the minimum recommended micronutrient intake. Make it measurable, clear and easily understandable for readers. Revised as commented 3-4

4. Use the updated national nutrition program policies and strategies of the country rather than using the outdated National Nutrition Program (NNP II) 2016-2020. Revised as commented 3-4

Introduction

1. Page 12: line 5-6: Use the sub title either of the word introduction or background. It is preferred to use the word introduction rather than the word background. It is more formal. . Revised as commented 4

2. Page 12: line 8-10: However, during this time, children are highly susceptible to undernutrition due to factors like inadequate dietary intake, limited access to food, nutritional taboos, and infectious diseases (2). The Authors list out the factors contributed for undernutrition with identifying the immediate cause of malnutrition. What about the underlining and basic causes that most commonly contribute for malnutrition? It would be better if the authors add those points with their citations. Revised as commented 4

Methods

Study design, setting and periods

1. Page 14: line 6: Afar National Regional State is one of the eleven regional states of Ethiopia. The Authors need to update it. Now corrected as commented 6-7

2. Page 14: line 10-13: The last administrative study was Districts and Kebeles. The Authors need to discuss districts available in each zone in detail. Revised. We only focused on Aysaita district of the zone considering comments number 9 attached in the document 6-9

3. Page 14: line 14-15: …and a total population projection for 2017 was 1,812,002, consisting of 991,000 men and 821,002 women (39). The Authors need to use the most recent population of the region. Revised as commented 6-7

4. Page 14: line 14-15: For this study, Assayta district was selected randomly, which was found in Awsi Rasu (Administrative Zone 1) within the Afar Region, Ethiopia. How one district (Assayta district-11 kebele) represents the entire Afar region? How would your study have generalized to the whole 6-23months children living in Afar region? Even how do you think that Assayta district findings will be generalized to Awsi Rasu (Administrative Zone 1)? In research, at least 30% of the target population should be sample to be generalized. The Authors should see this as a serious issue and your research title would be revised. We have revised the manuscript as suggested. We agree with your evidence-based comment regarding the need for a sample size representing 30% of the population for generalizability. To meet this requirement, we narrowed the study's scope to focus exclusively on Aysaita district. Consequently, the analysis and discussion are now specific to the district level. These changes are reflected throughout the manuscript, beginning with the title. 6-9 Thank you very much

5. Page 14: line 4-23: The study was conducted in pastoralist communities of Afar Region. However, I can’t see anything explained about the pastoralist communities. Does all the population of the Afar region totally (100%) pastoralist communities? Make it clear for the readers. Now revised as commented 6-7

6. Page 14: line 20-22: The district is administratively divided into 11 kebeles. According to the Aysaita District Administration, the total human population of the district is 61961. Among the total population of the district, 12392 households have children aged 6–23 months. Cite your reference here. Cited as commented. 6-7

7. Page 14: line 27-29: Source population, all children aged 6-23 months and mothers/caregivers residing in pastoralist communities of the Afar region were source populations. Did the mothers/caregivers were also your study population? Now revised as commented. The study population were all children aged 6-23 months.

8. Page 15: line 1-2: Your study population was All children 6-23 months old and who are residents of the selected kebeles in the selected district. How those children 6-23 months old systematically selected represent all children 6-23 months old residing in Afar region? Of course, not possible. We agree with your evidence-based comment regarding the need for a sample size representing 30% of the population for generalizability. To meet this requirement, we narrowed the study's scope to focus exclusively on Aysaita district. Consequently, the analysis and discussion are now specific to the district level. These changes are reflected throughout the manuscript, beginning with the title. 6-8 Thank you

9. Page 15: line 1-2: In your Exclusion criteria, Children living with others than their mothers or caregivers were excluded from the analysis. I think children living with others than their mothers or caregivers were in appropriately excluded from the study. Because, children living with others than their mothers or caregivers would be one factor for Minimum Recommended Micronutrient Intake. Now revised as commented. 6-7

10. Page 15: line 13-14: Why you used the OpenEpi online software version 3.1 instead of using population determination formula? It could be better if you were used the OpenEpi online software for determining the sample size for factors associated with the Minimum Recommended Micronutrient Intake status. We utilized the OpenEpi online software version 3.1 for sample size calculation to ensure accuracy, efficiency, and methodological rigor. While the standard population determination formula provides the theoretical foundation, manual computation is prone to arithmetic error and omits crucial adjustments. Additionally, the calculation became the same. However, the software calculation saves time, accurate and prevent calculation error. There was also typo error during the sample size calculation. That was also corrected. 7-9

11. Page 15: line 13-19: I didn’t see the sample size determination for factors associated with the Minimum Recommended Micronutrient Intake status. How do you think that the dependent variable sample size was representing the factors for your dependent variable? Of course, we have calculated already during proposal development. Now the detail calculated sample size is provided in the attached document for transparency. 7-9

12. Page 15: line 15: First, Zone One was randomly selected from the six zones in the Afar region. How this one zone will represent the other 5 zones/6 zones? It is a major issue to be solved. Now revised as commented 7-9

13. Page 17: line 15: MN. Always use the first abbreviations together with full phrase. Now revised as commented 10-11

Dissemination of result

1. Page 19-20: line 1-5: Remove all dissemination of the result. It is not the part of the article rather than during your research proposal. Removed as commented

Results

1. Page 21: Table 1.: Family size, what is your evidence for the categorizing Family size with ≤ 4 and >4 only? Now corrected as commented. We have updated our approach and now we used the average family size. 13

2. Page 21: Table 1.: Monthly income, does this income per month or annum? what is the currency of income? Which one is correct? USD or ETB? Much better and easily understandable if explained in USD for general readers but not only for Ethiopians. Revised as commented. We converted using current exchange to United States Dollar. 13-14

3. Page 21: line 31-32: It says…Of these, the majority of them, 580 (85.93%) belonged to the 12–18 age range. The age range written here-12-18 is quite different from the age group categorized in table 1, page 21: line 21-23. Please take corrective actions. Now revised as commented 14-17

4. Page 24: line 15: Relace the word “Again” with in “addition” Revised as commented

5. Page 26: The monthly income category in table 1 and 5 are quite different. Therefore, please update it. Revised as commented

6. Page 26: Table 5, the Asterisk (*) listed out in table 5 is not explained in foot note. Revised as commented 18

Discussion

1. Page 26: line 20-21: The possible reasons should be cited from credible studies. And try to cite research evidence of why these variables are associated, not just supposition. The comment works throughout the discussion. Now revised as commented 18-20

2. Page 26: line 16-21: Provide your reasons individually why your research finding was higher than X studies and why your research finding were lower than X studies based on relevant references/studies/citation. It works for all discussions. Revised as commented 18-20

3. Page 27: line 3-5: It that explanation of evidence or your recommendation. If it is your recommendation, please put it in Conclusion and recommendations component of your article. Now corrected as commented 18-21

Reference

1. Page 31-34: Use the selected/used references in your article other useless references should be removed. Now revised as commented

Annex

1. Add the questionaries used for data collection as a supplementary material. Well noted and uploaded as commented

Overall comment

1. This study is well-conducted, and the researchers are commended for their work thorough the title of Minimum Recommended Micronutrient Intake Status and Associated Factors among Children Aged 6-23 Months in Pastoralist Community of Afar Region, Ethiopia, 2024: Community Based Cross-sectional Study. The study is effectively structured, with a clear identification of the impact of Minimum Recommended Micronutrient Intake Status with well thought recommendations for public health interventions. However, to enhance its credibility and ensure clear communication of the findings, the researchers should address the minor comments and suggestions provided above. Once these revisions are made, the study will be ready for publication and will provide valuable insights for controlling Micronutrient deficiency in your study area and the country as well. Hope all the minor comments revised.

The authors are grateful for the appreciation and constructive comments on the manuscript you provided. I, as corresponding author found the detailed revisions incredibly helpful and educational. Addressing each point has significantly improved the quality of the work. Thank you for your time and guidance; we hope this revised version meets with approval for publication.

5. Author Response to Editor and Journal Requirement

Editorial or Journal Requirement: Comments

Author Response Page number in the manuscript

After careful consideration, we feel that it has merit but does not fully meet PLOS ONE’s publication criteria as it currently stands. Therefore, we invite you to submit a revised version of the manuscript that addresses the points raised during the review process. First of all, we would like to thank you for your encouragement, carefully assessment, constructive comments and request for revision

1. Please ensure that your manuscript meets PLOS ONE's style requirements, including those for file naming. Revised as commented All pages

2. In the ethics statement in the Methods, you have specified that verbal consent was obtained. Please provide additional details regarding how this consent was documented and witnessed, and state whether this was approved by the IRB. Detail explanation provided as commented. 12-13

3. In the online submission form, you indicated that “All data are available base on the reasonable request from principal investigator or corresponding author”. All PLOS journals now require all data underlying the findings described in their manuscript to be freely available to other researchers, either 1. In a public repository, 2. Within the manuscript itself, or 3. Uploaded as supplementary information. Data uploaded as supplementary information following comment provided.

4. Your ethics statement should only appear in the Methods section of your manuscript. If your ethics statement is written in any section besides the Methods, please move it to the Methods section and delete it from any other section. Please ensure that your ethics statement is included in your manuscript, as the ethics statement entered into the online submission form will not be published alongside your manuscript. Now moved to method section and removed in other part of the manuscript to avoid repetition

5. Please upload a copy of Figure 3, to which you refer in your text on page 23. If the figure is no longer to be included as part of the submission please remove all reference to it within the text. Removed as commented.

6. Please include captions for your Supporting Information fi

---

## [Decision Letter · Decision Letter 1]

1 Oct 2025

Minimum recommended micronutrient intake status and associated factors among pastoralist children aged 6-23 months in Aysaita district, Afar Region, Ethiopia, 2024. A community based cross-sectional study

PONE-D-24-55242R1

Dear Dr. Moloro,

We’re pleased to inform you that your manuscript has been judged scientifically suitable for publication and will be formally accepted for publication once it meets all outstanding technical requirements.

Kind regards,

Dinaol Abdissa Fufa, Mph

Academic Editor

PLOS ONE

Reviewers' comments:

Reviewer's Responses to Questions

**Comments to the Author**

Reviewer #1: All comments have been addressed

Reviewer #3: All comments have been addressed

Reviewer #4: All comments have been addressed

2. Is the manuscript technically sound, and do the data support the conclusions?

Reviewer #1: Yes

Reviewer #3: Yes

Reviewer #4: Yes

3. Has the statistical analysis been performed appropriately and rigorously?

Reviewer #1: Yes

Reviewer #3: Yes

Reviewer #4: Yes

4. Have the authors made all data underlying the findings in their manuscript fully available?

Reviewer #1: Yes

Reviewer #3: Yes

Reviewer #4: Yes

5. Is the manuscript presented in an intelligible fashion and written in standard English?

Reviewer #1: Yes

Reviewer #3: Yes

Reviewer #4: Yes

Reviewer #1: All comments have been addressed. Thank you for your effort and good luck in your future researches.

Reviewer #3: (No Response)

Reviewer #4: The research is well structured and done. All my comments were addressed. The supporting documments were also attached.

**Do you want your identity to be public for this peer review?** For information about this choice, including consent withdrawal, please see our Privacy Policy

Reviewer #1: No

Reviewer #3: No

Reviewer #4: **Yes: ** First name: Habtamu Molla

Last name: Ayele

Affiliation: Maternal and Child Health Directorate, Federal Ministry of Health, Addis Ababa, Ethiopia

---

## [Editor Report · Acceptance letter]

PONE-D-24-55242R1

PLOS ONE

Dear Dr. Moloro,

I'm pleased to inform you that your manuscript has been deemed suitable for publication in PLOS ONE. Congratulations! Your manuscript is now being handed over to our production team.

Kind regards,

on behalf of

Dr. Dinaol Abdissa Fufa

Academic Editor

PLOS ONE